# Goniotomy for Non-Infectious Uveitic Glaucoma in Children

**DOI:** 10.3390/jcm12062200

**Published:** 2023-03-12

**Authors:** Charlotte L. L. I. van Meerwijk, Astrid B. Edema, Laurentius J. van Rijn, Leonoor I. Los, Nomdo M. Jansonius

**Affiliations:** 1Department of Ophthalmology, University Medical Center Groningen, P.O. Box 30.001, 9700 RB Groningen, The Netherlands; a.b.edema@umcg.nl (A.B.E.); l.i.los@umcg.nl (L.I.L.); n.m.jansonius@umcg.nl (N.M.J.); 2Department of Ophthalmology, Amsterdam University Medical Center, location VU University Medical Center, P.O. Box 7057, 1007 MB Amsterdam, The Netherlands; vanrijn@amsterdamumc.nl

**Keywords:** glaucoma, surgery, uveitis, childhood

## Abstract

Secondary glaucoma is still a blinding complication in childhood uveitis, for which most commonly used surgical interventions (trabeculectomy or glaucoma drainage implant) involve multiple re-interventions and/or complications postoperatively. The goniotomy procedure has never been investigated in the current era, in which patients with pediatric uveitis receive biologics as immunosuppressive therapy for a prolonged period, with potential implications for the outcome. The purpose of the study is to evaluate the efficacy and safety of a goniotomy procedure in pediatric non-infectious uveitis in a retrospective, multicenter case series. The primary outcomes were the postoperative intraocular pressure (IOP), number of IOP-lowering medications, and success rate. Postoperative success was defined as 6 ≤ IOP ≤ 21 mmHg, without major complications or re-interventions. Fifteen eyes of ten children were included. Median age of the included patients at goniotomy was 7 years; median follow-up was 59 months. Median (interquartile range) IOP before surgery was 30 (26–34) mmHg with 4 (3–4) IOP-lowering medications. At 1, 2, and 5 years after goniotomy, median IOP was 15, 14, and 15 mmHg with 2 (0–2), 1 (0–2), and 0 (0–2) medications, respectively (*p* < 0.001 postoperatively versus preoperatively for all timepoints). Success rate was 100%, 93%, and 80% after 1, 2, and 5 years, respectively. There were no significant changes in visual acuity and uveitis activity or its treatment, and there were no major complications. Our results show that the goniotomy is an effective and safe surgery for children with uveitic glaucoma.

## 1. Introduction

Children ≤16 years of age constitute 5–10% of the uveitis population [1]. The incidence of uveitis in children is 4–6 per 100,000 person-years [2,3], of which a non-infectious etiology is the most common. Non-infectious pediatric uveitis is a challenge to treat, because of its often-asymptomatic presentation, ocular evaluation difficulties, the high impact of treatment, the chronic course, and the risk of serious complications.

Secondary glaucoma is one of the most common complications, together with cataract, optic disc edema, and cystoid macular edema [4]. Moreover, glaucoma is nonreversible and thereby potentially blinding [5]. Based on a retrospective analysis in a tertiary care referral center, secondary glaucoma developed in 26% of the children with uveitis. In 58% of these children, the required intraocular pressure (IOP) reduction was not achieved with IOP-lowering medication only, and additional surgical interventions were needed in order to prevent or limit optic nerve damage [6].

The pathophysiology of uveitic glaucoma is multifactorial [7]. Typically, there is mechanical obstruction or dysfunction of the trabecular meshwork, which can be blocked by inflammatory cells, proteins, and debris liberated from a disrupted blood-aqueous barrier. In addition, alterations in the trabecular meshwork due to a reaction to steroids may occur. All these factors may result in an obstruction of the aqueous outflow facility [8]. In chronic uveitis, obstruction of aqueous outflow may result from scarring and obliteration of trabecular meshwork beams or Schlemm’s canal or from overgrowth of a fibrovascular membrane in the chamber angle [8].

In order to achieve an IOP decrease in this type of secondary glaucoma, various surgical interventions are possible. The most frequently used techniques are trabeculectomy (TE) and implantation of a glaucoma drainage device. Another option is goniotomy. Previous research and a recent review showed no preference for a particular intervention [9,10]. Advantages of goniotomy over the other two types of intervention are a quick surgical procedure and a relatively rapid postoperative recovery.

The goal of goniotomy is to facilitate the entrance of aqueous humor into Schlemm’s canal. Incising the trabecular meshwork is presumed to lower aqueous outflow resistance, leading to improved IOP control [11]. Histopathological studies showed a superficial, non-healing incision, with entrance to Schlemm’s canal [12]. During goniotomy, the anterior trabecular meshwork is incised just below Schwalbe’s line by using a knife or needle under direct visualization of the chamber angle with a gonioscopy lens. Goniotomy is usually performed initially over 4 to 6 clock hours in the nasal, temporal or inferior angle. The procedure can be repeated (extended to more clock hours) in case of an inadequate IOP-lowering effect [13].

Three studies have been published on goniotomy procedures in pediatric uveitis [14,15,16]. They all showed a significant decrease in IOP and number of IOP-lowering medications, although multiple interventions were often needed. Since these studies were published, however, uveitis treatment has undergone a tremendous development, with stricter control of uveitis activity, early use of systemic disease-modifying antirheumatic drugs (DMARDs), and strict monitoring and early treatment when the IOP rises. Nevertheless, despite these developments, pediatric uveitic glaucoma has not found a gold standard treatment yet. 

The aim of the current study was to evaluate the efficacy and safety of goniotomy in pediatric uveitis in an era with a paradigm shift regarding uveitis treatment. For this purpose, we performed a retrospective, multicenter case series study involving all patients from tertiary centers in The Netherlands, where goniotomy is the current approach to treat glaucoma in children with uveitis.

## 2. Materials & Methods

### 2.1. Study Population

Patients with non-infectious pediatric uveitis, who underwent a goniotomy procedure from December 2011 until March 2020, were retrospectively included from the departments of Ophthalmology of the University Medical Center (UMC) Groningen (Center 1) and the Amsterdam UMC (Center 2), both in The Netherlands. The diagnosis and classification of uveitis was done by ophthalmologists specialized in pediatric uveitis and according to the Standardization of Uveitis Nomenclature (SUN) criteria [16]. Children were evaluated for the presence of an underlying systemic disease by pediatric rheumatologists. The indication for glaucoma surgery was made by glaucoma specialists and was mostly based on an IOP > 21 mmHg with maximum tolerated IOP-lowering medication and individual patient characteristics. The included goniotomy procedures were done as the first surgical intervention for glaucoma, with a minimal follow-up after the intervention of 1 year. If bilateral goniotomy was performed, both eyes of the patient were included (see Section 2.5 ‘Statistical analysis’). The study adhered to the tenets of the Declaration of Helsinki. Because the study concerns a retrospective analysis of data that have been collected during regular patient care, no formal approval of a Medical Ethical Committee was required (waiver obtained from the Medical Ethical Committee of both participating centers).

### 2.2. Technique

A standard goniotomy procedure as described by Worst was used with very little modifications [17]. During the procedure, the anterior trabecular meshwork was incised just below Schwalbe’s line by using a knife/needle under direct visualization of the chamber angle with a gonioscopy lens. Usually, 4 to 5 clock hours were treated. After the procedure, a soluble suture was placed at the corneal incision. After surgery, the number of topical steroid administrations per day was increased temporarily. In addition, in Center 1, methylprednisolone 15 mg/kg was given during surgery. The DMARD use remained unchanged in both centers.

### 2.3. Data Collection

Preoperative data extracted from the patient’s electronic medical records included: age at surgery, sex, type of uveitis, etiology of uveitis, antinuclear antibody (ANA) seropositivity, anterior complications at first presentation (peripheral anterior synechiae (PAS), cataract, band keratopathy), posterior complications at first presentation (cystoid macular edema, vasculitis, papillitis), lens status at last visit before surgery, interval between onset of uveitis and surgery, and interval between start of IOP-lowering medication and surgery. Onset of uveitis was defined as the first time uveitis was diagnosed, either in the included centers or elsewhere as specified in the referral letter. 

The following information was recorded at the last visit before surgery and every year (±3 months) after surgery up to 5 years of follow-up: IOP, number of different classes of IOP-lowering medication used, use of DMARDs, best-corrected visual acuity, anterior chamber inflammation activity according to the SUN grading system for anterior chamber cells [18], and use of topical steroid medication. End of follow-up was defined as the date of either the last visit in 2022 or failure (as defined below). Complications and additional interventions during follow-up were recorded. The diagnosis of cystoid macular edema was made, when any accumulation of fluid in the macular area was seen by spectral domain optical coherence tomography. Papillitis was defined as blurring of the optic disc margins visible by fundoscopy and/or the presence of optic disc hyperfluorescence on fluorescein angiography. The IOP was measured by Goldmann Applanation Tonometry when possible. Icare rebound tonometer readings were used if the child was unable to cooperate with Goldmann Applanation Tonometry. As systemic anti-inflammatory medication, we scored systemic steroids (excluding short-term perioperative steroid treatment), conventional synthetic (cs)DMARDs such as methotrexate and mycophenolate mofetil, and biologic (b)DMARDs such as adalimumab and infliximab (anti-TNF-alpha).

### 2.4. Study Outcomes

Primary outcomes were IOP, number of IOP-lowering medication classes after surgery, and success or failure after surgery. Success was defined as no failure regardless of the use of IOP-lowering medication. Failure was defined as any of the following: (1) an IOP > 21 or <6 mmHg on at least 3 consecutive occasions from 3 months after the goniotomy onwards; (2) an IOP <10 mmHg together with a hypotonic maculopathy, optic disc edema, vision loss or choroidal detachment; (3) the need for additional glaucoma-related surgical interventions; (4) loss of light perception. We defined vision loss as a visual acuity loss of more than 2 Snellen chart lines. Secondary outcomes included complications, uveitis activity, and visual acuity.

### 2.5. Statistical Analysis

For descriptive statistics, we used median and interquartile range (IQR), because of the non-normal distribution of the data. Medians were compared using the Wilcoxon signed rank test (for paired samples) and the Wilcoxon rank sum test (for independent samples). For nominal data, we used the McNemar’s test for paired samples and the Pearson chi-square test for independent samples. The effect of goniotomy on IOP and on the number of IOP-lowering medication classes was studied with (generalized) linear mixed-effects models, in order to be able to account for the inclusion of both eyes in some patients and for the assessment of IOP and the number of medication classes at multiple timepoints. Dependent variable was either IOP or the number of medication classes; timepoint was entered in the models as a fixed effect and factorized (with 6 levels: preoperatively and 1, 2, 3, 4, and 5 years postoperatively), with the preoperative time point as reference. Center, patient, and eye were entered as nested random effects. As a secondary analysis, we entered ‘center’ as a fixed effect, to study if there were any differences in the effect of goniotomy on IOP between the centers. A Kaplan–Meier curve was used to illustrate at which time points during the follow-up failures occurred.

Data were statistically analyzed with SPSS 28.0.0 (SPSS Inc., Chicago, IL, USA). For the (generalized) linear mixed-effects models, we used package ‘lme4’ in R (version 4.2.2; R Foundation for Statistical Computing, Vienna, Austria) with ‘lmer’ for IOP and ‘glmer’ with family = poisson for the number of IOP-lowering medication classes. A *p*-value of 0.05 or less was considered statistically significant. 

## 3. Results

A total of 15 eyes of 10 patients were included; 8 eyes in Center 1 and 7 eyes in Center 2. Table 1 gives an overview of the demographics. Median age at goniotomy was 7 (IQR 6–11) years and median follow-up after surgery was 59 (37–83) months. The included eyes were most often of female patients, and the uveitis was mainly JIA-related and anteriorly located. In 47% of eyes, pre-existing anterior complications were seen at the first visit in the expert center. Three eyes had had cataract surgery with IOL implantation before their goniotomy procedure, and one eye had PAS in a limited number of clock hours, without clear progress and not involving the goniotomy site. None of the patients were lost to follow-up.

Figure 1 and Table 2 show the results for IOP and the number of IOP-lowering medication classes. During follow-up, a significant decrease in IOP and the number of IOP-lowering medication classes was seen at every postoperative time point as compared to the preoperative time point (*p* < 0.001 for both IOP and the number of IOP-lowering medication classes at all timepoints). The greatest reduction of IOP-lowering medication was 1 year after goniotomy, with 7 eyes of 4 patients not receiving any IOP-lowering medication from this moment onwards. In one patient with two eyes, the amount of IOP-lowering medication was slowly tempered over time, ending without IOP-lowering medication from year 4 onwards. Three patients with three eyes included use of two or three types of IOP-lowering medication throughout the postoperative period without tempering medication over time. There was a small difference between the centers regarding the effect of goniotomy on IOP (*p* = 0.047), mainly related to a somewhat higher preoperative IOP in Center 1 (median preoperative IOP 33 versus 26 mmHg for Center 1 versus Center 2, respectively). With regard to visual acuity, uveitis activity, the frequency of topical steroid medication, and use of DMARDs, no clear differences were found between the preoperative and postoperative time points (Table 2).

Figure 2 presents a Kaplan–Meier curve. The success rates were 100% (*n* = 15/15), 93% (*n* = 14/15), and 80% (*n* = 8/10), after 1, 2, and 5 years, respectively. The two failures concerned eyes from different patients who underwent additional glaucoma surgery; one eye received a Baerveldt drainage implant 16 months after the goniotomy procedure, because of periods of IOP increase due to low compliance with a glaucomatous excavation of the optic disc; one eye received a TE after 28 months, because of a high IOP with the maximum IOP-lowering medication.

None of the eyes had major adverse events postoperatively. In one eye a macroscopic hyphema occurred after goniotomy with spontaneous resolution in one week. Three eyes had a mild uveitic reaction after goniotomy that endured for 3 and 8 months, respectively, and persisted in one eye (in a patient without DMARDs). Three eyes had a transient high IOP post-operatively. Two eyes underwent cataract surgery after goniotomy (after 2 and 16 months, respectively); both of them were already diagnosed with cataract before the glaucoma intervention. 

## 4. Discussion

Goniotomy reveals a significant decrease in IOP and the number of IOP-lowering medication classes in non-infectious uveitic glaucoma in children; in our data, the decrease remained stable during a 5-year follow-up. No differences between pre- and postoperative data were found with regard to visual acuity, uveitis activity, frequency of topical steroid medication, and use of different systemic medications, suggesting a safe procedure and a stable uveitis over time.

Previous studies showed similar significant decreases in IOP and number of IOP-lowering medication classes after surgery, but showed a lower success rate after one goniotomy; 72–76% [14,15] after 1 year, 54–61% [14,16] after 2 years, and 47–54% [14,16] after ≥5 years of follow-up. While in earlier studies PAS, aphakia, and multiple previous ocular surgeries were described as predictors of surgical failure, in our study we had only one eye with limited PAS, no aphakic eyes, and no eyes with multiple previous ocular surgeries. It is probable that the more fortunate baseline and preoperative findings in our cohort contributed to a better outcome post-operatively. In the eye with PAS, the areas with synechiae were avoided during the goniotomy procedure and this eye had a successful outcome with no IOP-lowering medication at the end of follow-up. 

We did not perform additional analyses to determine potential risk factors for risk of failure because we had just two failures in our cohort. However, the risk factors for failure published by previous studies correspond with the characteristics of our failures. For example, Ho and Walton [19] found a higher risk of failure in cases with a longer interval between glaucoma diagnosis and surgery (mean 1.1 ± 1.0 years in their success versus 5.1 ± 3.3 years in the failure group, *p* = 0.001). In our cohort, the failures had a longer interval between the start of IOP-lowering medication and surgery as well (16 and 28 months in the failures versus 9 (6–19) months (median (IQR)) in the success group). Additionally, Ho and Walton [19] published a significantly higher age at surgery in the failure group (mean 9.2 ± 3.7 years in their success versus 12.6 ± 4.7 years in their failure group, *p* = 0.04). A difference in age at surgery was also present in our cohort (11 and 13 years in the failures versus 7 (6–10) years (median (IQR)) in the success group). These observations support the advice to perform prompt glaucoma surgery when it is indicated. 

Adequate uveitis control in non-infectious pediatric uveitis is of great importance, and an inactive uveitis for at least 3 months before surgery is recommended [7,20]. In order to reach an inactive uveitis, DMARDs were often used preoperatively; csDMARDs in nine patients (90%) and bDMARDs in five patients (50%). None of the patients used systemic steroids for a prolonged period of time. Previous studies also describe the use of csDMARDs and systemic steroids [14,16]. Bohnsack et al. [15]. was the only study that reported more details: 69% of the patients used DMARDs (with no differentiation between csDMARDs and bDMARDs) and 14% of the patients used systemic steroids. 

The working mechanism of the goniotomy in uveitic glaucoma is not completely understood. Most likely, incising the damaged trabecular meshwork results in a better aqueous humor outflow, leading to improved IOP control [11]. In a histopathological study after a trabeculodialysis in uveitic glaucoma, the treated meshwork was scarred, with adjacent collagen cores cemented together with loss of endothelial covering and absence of intertrabecular spaces and thereby collapsed structures. However, it is not only the connection with or entrance to Schlemm’s canal that seems to influence the outflow of the aqueous humor via the trabecular meshwork pathway. The pulsation mechanism to flush aqueous efficiently to Schlemm’s canal and into the draining venous system, activated by stretching of the trabecular meshwork, secondary to IOP rise, is also important for IOP regulation [21]. It is plausible that the fibrosing process, due to chronic inflammation, decreases the ability to pulsate. In addition, pro-inflammatory factors are produced during an IOP increase [22]. Thus, early control of uveitis activity and early surgical treatment of medically uncontrolled elevated IOP could both have a protective effect on the structure of the trabecular meshwork and contribute to the more favorable outcomes of goniotomy surgery now as compared to the past. A major advantage of the goniotomy is the ability to continue DMARD, as the survival of filtering blebs and the occurrence of fibrosis, such as after a glaucoma drainage implant, is not a factor [23,24,25].

In our study and in the three previous studies about the goniotomy procedure in pediatric uveitis [14,15,16], exacerbation of uveitis did not occur and no major complications such as infection, iatrogenic damage of intraocular structures or hypotony occurred. A small, transient hyphema, macroscopic but without the need of any intervention, occurred in our cohort. Blood visible in the anterior chamber angle is considered a normal phenomenon after goniotomy, presumed to be caused by reflux and considered a sign of success. Two out of twelve phakic eyes (17%) underwent cataract surgery after goniotomy; both of them were already diagnosed with cataract before the glaucoma intervention. In previous research [14,15,16], 10–44% of the phakic eyes, which had some lens opacity before glaucoma surgery, needed cataract surgery after a goniotomy procedure. Cataract progressed in only a few cases after goniotomy. The complication profile of the goniotomy procedure is mild when compared to that of TE and glaucoma drainage implants, which have a risk of hypotony and bleb or implant-based complications [10].

In spite of its retrospective design, our study does not suffer from selection bias, due to the fact that goniotomy was the first step in the surgical protocol during the inclusion period (except in one patient, where the first treated eye had failed). In all patients, visualization and accessibility of the chamber angle were sufficient to perform a goniotomy. Goniotomy procedures were performed by two surgeons (one surgeon per center). The relatively long follow-up period and the reporting of data at fixed time points are other strengths of our study. Potential weaknesses of our study include its limited sample size and a lack of predefined standardization in measurements and time points. Success was predominantly based on IOP outcomes. Ideally, visual field outcomes should have been added, but obtaining reliable visual field outcomes in young children is challenging. In addition, the two eyes in which goniotomy failed were treated directly with other forms of glaucoma surgery; therefore, it is unclear whether additional goniotomy would have had an added benefit. 

Further research regarding goniotomy in pediatric uveitic glaucoma is warranted, preferably using prospective randomized controlled studies with a larger sample size that compare different types of glaucoma surgery, especially as a function of age.

## 5. Conclusions

This study indicates that goniotomy might be considered as a primary surgical treatment in children with non-infectious uveitic glaucoma. Goniotomy is a safe and straightforward procedure, and it appears to be effective over a relatively long follow-up period. It might be a definite treatment in the majority of eyes, and it can be supplemented by more extensive glaucoma surgery at a later time point if necessary.

## Figures and Tables

**Figure 1 jcm-12-02200-f001:**
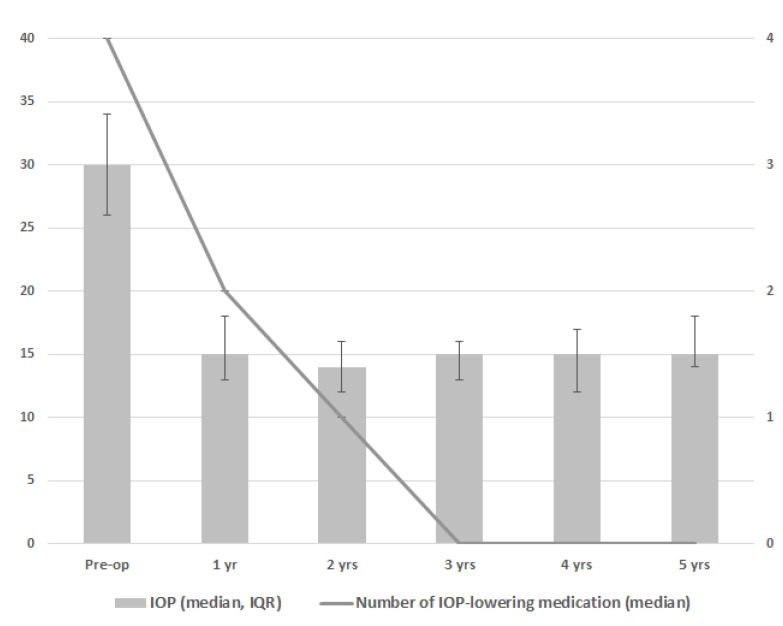
Median intraocular pressure (IOP; gray bars) and median number of IOP-lowering medications (line) after one goniotomy procedure based on 15 eyes in 10 patients. Error bars depict interquartile range of the IOP.

**Figure 2 jcm-12-02200-f002:**
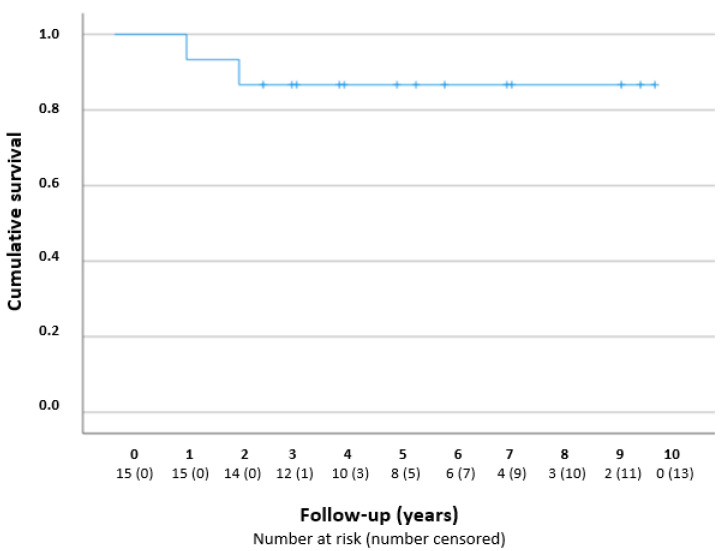
Kaplan–Meier curve—cumulative survival after one goniotomy procedure in years. The number at risk is reported per year during the follow-up.

**Table 1 jcm-12-02200-t001:** Demographics per eye.

Demographics	
Number of patients	10
Number of eyes	15
Sex—Female	13 (87%)
Age at goniotomy (yrs) ^a^	7 (6–11)
Localization—anterior uveitis	13 (87%)
Etiology—JIA ^b^	9 (64%)
ANA positivity ^c^	9 (64%)
Pre-existing complications at first visit UMC	
Anterior complications—yes	7 (47%)
Peripheral anterior synechiae—yes	1 (7%)
Posterior synechiae—yes	6 (40%)
Band keratopathy—yes	3 (20%)
Cataract—yes	0 (0%)
Posterior complications ^d^—yes	4 (27%)
Lens status before goniotomy ^e^	
Phakic eyes—no cataract	9 (60%)
Phakic eyes—cataract	3 (20%)
Pseudophakic eyes	3 (20%)
Interval (months) ^a^ between	
Onset of uveitis and goniotomy	22 (9–56)
Start glaucoma medication and goniotomy	9 (6–19)
Postoperative follow-up (months) ^a^	59 (37–83)

^a^ Median (interquartile range), ^b^ Juvenile Idiopathic Arthritis, ^c^ Antinuclear Antibodies, ^d^ Cystoid macular edema, vasculitis and/or papillitis, ^e^ No cataract surgery was performed in the period between the first presentation and goniotomy.

**Table 2 jcm-12-02200-t002:** Primary and secondary outcomes during the follow-up.

Characteristics	Pre-Operatively	1 Year	2 Years	3 Years	4 Years	5 Years
N (patients/eyes)	10/15	10/15	9/14	7/12	6/10	5/8
IOP (mmHg) ^a,b^	30 (26–34)	15 (13–18)	14 (12–16)	15 (13–16)	15 (12–17)	15 (14–18)
IOP lowering medication ^a,b,c^	4 (3–4)	2 (0–2)	1 (0–2)	0 (0–2)	0 (0–1)	0 (0–2)
AC inflammation ^b,d^	0 (0–1)	0 (0–1)	0 (0–1)	0 (0–0)	0 (0–1)	0 (0–0)
Topical steroid medication ^b,e^	2 (1–3)	1 (1–3)	1 (1–3)	1 (1–2)	2 (1–2)	1 (1–2)
Visual acuity (logMAR) ^b^	0.05 (−0.03–0.30)	0.10 (0.00–0.13)	0.00 (−0.03–0.03)	−0.05 (−0.10–0.18)	0.00 (−0.10–0.20)	0.00 (−0.10–0.18)
csDMARDS (N pt/eyes) ^f^	9/14	9/14	8/13	6/10	3/6	2/4
bDMARDS (N pt/eyes) ^g^	5/8	6/10	6/10	4/8	3/6	3/6

^a^ Intraocular pressure, ^b^ Median (interquartile range) per eye, ^c^ Number of different classes of IOP-lowering medication, ^d^ Anterior chamber inflammation, graded according to the Standardization of Uveitis Nomenclature (SUN) guidelines (Jabs et al.), ^e^ In units; prednisolone drops and ointment and dexamethasone drops were scored as 1 unit per drop. Rimexolone (Vexol, Alcon bv) and fluorometholone (FML liquifilm, Abbvie bv) were scored as 0.5 units per drop. No subconjunctival triamcinolone was used, ^f^ Conventional synthetic disease-modifying antirheumatic drugs, ^g^ Biologic disease-modifying antirheumatic drugs.

## Data Availability

Not applicable.

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
