# Peer review of "Goniotomy for Non-Infectious Uveitic Glaucoma in Children"

_jcm, 2023, doi:10.3390/jcm12062200_

Round 1

Reviewer 1 Report

The study is interesting and the manuscript well written. Weakness are mentioned in discussion. 

Although with a small sample size (but significant as the condition is relatively uncommon) the paper reports interesting information about a surgical technique for the management of a potentially blinding condition. The follow-up is also significant. 

I fell that the result of this study can add something to the current knowledge about the management of uveitic paediatric glaucoma 

Author Response

Dear reviewer,

We would like to thank the reviewer for the positive response on our manuscript. Since we did not receive any points of improvement, we would like to thank the reviewer for the effort in reviewing our study and sharing expertise with us. 

Best regards,

Charlotte van Meerwijk

Reviewer 2 Report

Line 21, 22 I presume that you mean “Post-surgical” IOP, meds and success rate but this is NOT stated and should be.

Line 127. Please give some detail about the OCT scans (slice or zone) and criterion for CME from OCT scans.

Line 141, define level of vision loss (eg. 5 lines?)

Table 2 shows that by 3 years 0 IOP lowering Meds were used compared with earlier years. This seems to imply improvement BUT in yr 3 only 7 patients were reviewed and in earlier years there were more. Please comment on whether these 7 patients were using Meds in years 1 and 2, likewise for years 4 and 5 where patient numbers are lower.

Table 2. Visual Acuity data is jumbled. Perhaps add median on 1 line and range underneath?

Lines 194 to 195. Can you please comment on the cause for failure in the two individuals – high/uncontrollable IOP with max meds?

Author Response

We would like to refer you to our attached revision letter.
